# Short-Term Exposure to Nitrogen Dioxide Modifies Genetic Predisposition in Blood Lipid and Fasting Plasma Glucose: A Pedigree-Based Study

**DOI:** 10.3390/biology12121470

**Published:** 2023-11-27

**Authors:** Huangda Guo, Mengying Wang, Ying Ye, Chunlan Huang, Siyue Wang, Hexiang Peng, Xueheng Wang, Meng Fan, Tianjiao Hou, Xiaoling Wu, Xiaoming Huang, Yansheng Yan, Kuicheng Zheng, Tao Wu, Liming Li

**Affiliations:** 1Department of Epidemiology and Biostatistics, School of Public Health, Peking University, Beijing 100191, China; 1710306131@pku.edu.cn (H.G.);; 2Key Laboratory of Epidemiology of Major Diseases (Peking University), Ministry of Education, Beijing 100191, China; 3Department of Nutrition and Food Hygiene, School of Public Health, Peking University, Beijing 100191, China; 4Fujian Provincial Center for Disease Control and Prevention, Fuzhou 350012, China; 5Department of Hygiene, Nanjing Country Center for Disease Control and Prevention, Nanjing 363600, China; 6Key Laboratory of Reproductive Health, Ministry of Health, Beijing 100191, China; 7Center for Public Health and Epidemic Preparedness & Response, Peking University, Beijing 100191, China

**Keywords:** pedigree-based cohort, genotype–environment interaction, nitrogen dioxide, blood lipid, fasting plasma glucose

## Abstract

**Simple Summary:**

This study aims to explore the effects of short-term exposure to nitrogen dioxide on blood lipids and fasting plasma glucose, as well as the interaction effects with genetic factors. This topic is important as it can reveal changes in gene expressions across different environmental levels, providing insights for subsequent analyses of loci-specific interaction effects and facilitating the interpretation of published studies on specific gene interactions. The results provide new evidence on the associations between nitrogen dioxide and blood lipids or fasting plasma glucose, which are crucial risk factors for cardiovascular disease and diabetes. In addition, we found a potential interaction between genotype and nitrogen dioxide in lipids and fasting plasma glucose, suggesting that future studies could prioritize nitrogen dioxide exposure if they can identify specific genetic variants behind the genotype–environment interactions that affect lipids and fasting plasma glucose.

**Abstract:**

(1) Background: Previous studies suggest that exposure to nitrogen dioxide (NO_2_) has a negative impact on health. But few studies have explored the association between NO_2_ and blood lipids or fasting plasma glucose (FPG), as well as gene–air pollution interactions. This study aims to fill this knowledge gap based on a pedigree cohort in southern China. (2) Methods: Employing a pedigree-based design, 1563 individuals from 452 families participated in this study. Serum levels of triglycerides (TG), total cholesterol (TC), low-density lipoprotein cholesterol (LDLC), high-density lipoprotein cholesterol (HDLC), and FPG were measured. We investigated the associations between short-term NO_2_ exposure and lipid profiles or FPG using linear mixed regression models. The genotype–environment interaction (GenoXE) for each trait was estimated using variance component models. (3) Results: NO_2_ was inversely associated with HDLC but directly associated with TG and FPG. The results showed that each 1 μg/m^3^ increase in NO_2_ on day lag0 corresponded to a 1.926% (95%CI: 1.428–2.421%) decrease in HDLC and a 1.400% (95%CI: 0.341–2.470%) increase in FPG. Moreover, we observed a significant genotype–NO_2_ interaction with HDLC and FPG. (4) Conclusion: This study highlighted the association between NO_2_ exposure and blood lipid profiles or FPG. Additionally, our investigation suggested the presence of genotype–NO_2_ interactions in HDLC and FPG, indicating potential loci-specific interaction effects. These findings have the potential to inform and enhance the interpretation of studies that are focused on specific gene–environment interactions.

## 1. Introduction

Air pollution poses a significant and pervasive threat to global public health, contributing to 7.6% of all fatalities in 2015 [1]. Notably, a substantial proportion of the air pollution-related deaths in China are attributable to cardiovascular diseases (CVD) [2]. Given that CVDs stand as the leading cause of mortality globally, accounting for 32% of all deaths worldwide [3], we need to enhance our understanding of the intricate relationship between air pollution and the common risk factors associated with CVD, which is the key to advancing strategies for prevention and treatment.

Previous research has found that lipid and blood glucose levels could explain the relationship between air pollution and CVD [4,5,6,7,8,9,10]. For instance, Cai et al. found that a one-quartile interval increase in PM_10_ was associated with a 1.9% increase in TG [4]. Another study based on the NHANES cohort also found that elevated PM_10_ concentrations were associated with raised TG and TC levels [5]. A longitudinal study from China found that PM_10_ was associated with elevated blood glucose concentrations [6]. All of this evidence suggests a possible link between air pollution and cardiometabolic disease risk [4]. On the other hand, NO_2_, a potent atmospheric pollutant strongly associated with CVD, is a brown/orange irritant radical gas present in the environment. Common environmental sources of NO_2_ include mobile emissions, fuel combustion, industrial processes, and fires [11,12]. One plausible explanation for the association between NO_2_ and CVD is that NO_2_ induces abnormalities in lipid [7,8] and glucose [6] metabolism. Extensive evidence from animal studies support this claim, demonstrating a connection between air pollution and disrupted lipid profiles and elevated blood glucose levels [13,14]. Nevertheless, the body of population-based evidence on this topic remains both insufficient and inconsistent [8]. Furthermore, while much of the research on air pollution and its impact on lipid and glucose metabolism primarily focuses on airborne particulate matter, relatively fewer studies delve into the effects of gaseous pollutants like NO_2_.

Genetic factors also play a crucial role in lipid and glucose metabolism. Previous studies have estimated the heritability of lipid profiles [15,16,17] and fasting plasma glucose (FPG) levels [18,19]. Neglecting genetic influences or the genetic diversity across different populations might be one of the potential reasons for the inconsistent findings when investigating the connection between air pollution and dyslipidemia or hyperglycemia [20,21]. It is vital to discern whether susceptibility to adverse environmental exposures is genetically determined, a concept known as gene–environment interactions [22], and to elucidate the specific nature of these interactions. Such insights can aid in stratifying populations for targeted interventions. Both population-based and family-based studies have revealed interactions between inhaled particles and specific genetic variants concerning blood lipid levels, blood pressure, and blood glucose [20,21,23,24,25]. However, there is a scarcity of research exploring the impact of gene–NO_2_ interactions on blood lipids and FPG.

On the other hand, before delving into gene–environment interactions, we need quantitative genetics to determine whether traits are influenced by genotype–environment interactions (GenoXE). This preliminary step, often overlooked in practice, is essential as it helps establish the groundwork for subsequent investigations centered on specific environmental exposures and genetic loci. Unfortunately, this is rarely pursued due to the lack of adequately characterized cohorts with genealogical data [26]. GenoXE pertains to the interaction of overall additive genetic effects with environmental factors, meaning that the overall genetic influence on individual phenotypes varies across different levels of environmental exposures [27]. Importantly, this approach can be executed without the need for genotyping data but with a pedigree-based design.

Accordingly, the present study was undertaken to investigate the associations between NO_2_ exposure and lipid profiles as well as FPG levels. Furthermore, our analysis was conducted using extended pedigrees, allowing us to identify genotype–NO_2_ interactions. This approach lays the foundation for subsequent investigations focused on analyzing interaction effects at specific genetic loci and aids in the interpretation of previously published studies that delve into locus-specific interactions.

## 2. Materials and Methods

### 2.1. Study Population

We gathered data during the baseline survey of the Fujian Tulou Pedigree-based Cohort Study (FTPC), which was conducted between August 2015 and December 2017. The Fujian Tulous represent traditional Chinese rural dwelling styles in southeastern coastal China. Residents in this area organize themselves independently based on their family lineages, with a single surname predominating the population living in Tulous of various architectural forms. These Tulou pedigrees are typically expansive and characterized by intricate kinship ties, which facilitates the recruitment of more than three generations of family members—a practice rarely observed in other regions of China. Detailed information on the FTPC was reported in previous study [28]. Ultimately, our study included 452 families, comprising a total of 1563 study subjects. The study was approved by the Medical Research Ethics Committee of the Peking University Health Science Center. Informed consent was obtained before their participation in the study. The study was conducted in accordance with the Declaration of Helsinki.

### 2.2. Air Pollutant Data

We acquired air pollutant data, including the 24 h average concentrations of NO_2_, particulate matter (PM_2_._5_ and PM_10_), carbon monoxide (CO), and ozone (O_3_), in Zhangzhou City, Fujian Province, spanning from 1 August 2015 to 31 December 2017. These data were sourced from the National Air Pollution Monitoring System. The national standard monitoring station is equipped with 24 h automatic monitoring instruments for NO_2_ concentrations. Daily average concentrations were computed for our analysis. It is noteworthy that the monitoring data have been demonstrated to be a reliable representation of population exposure to air pollution in China [29,30]. A map illustrating the site of the Fujian Tulou Pedigree-based Cohort study and the locations of air pollutant monitors can be found in the Appendix A online, Appendix A.

### 2.3. Measurements

All participants underwent in-person interviews, physical examinations, and biochemical measurement by trained and certified investigators. A structured questionnaire was applied to collect basic demographic characteristics (age, sex, education), lifestyle information (smoking, alcohol drinking, physical activity, vegetable and fruit intake, whole-grain intake, and meat intake), and medication history (antihypertensive drugs, antidiabetic drugs, antihyperlipidemic drugs). Anthropometric variables, including height (cm) and weight (kg), were measured by trained technicians according to standard procedures. Body mass index (BMI) was then calculated as weight (kg) divided by squared height (m^2^). Overnight fasting venous blood (at least 8 h) was collected from participants for measuring their blood lipid levels, including TC, TG, LDLC, and HDLC, as well as their FPG levels. Blood specimens were processed and stored in Eppendorf tubes at the examination center, and sera were stored at −80 °C.

### 2.4. Statistical Analysis

We investigated the short-term relationships between air pollutants and lipid profiles or fasting plasma glucose (FPG) using four linear mixed models, each with distinct sets of adjusted covariates.

In Model 1, we controlled for age, sex, body mass index (BMI), and education level. Additionally, due to the pedigree-based design, we introduced family ID as a random effect in the model to account for correlations among individuals within families. Model 2 extended the adjustments to include factors such as smoking, alcohol consumption, physical activity, and dietary habits, encompassing the intake of fresh fruits and vegetables, whole grains, and meat, and tea consumption. In Model 3, we expanded our adjustments to encompass medication history, building upon the covariates considered in Model 2. In Model 4, we further incorporated the concentrations of particulate matter (PM_2_._5_ and PM_10_), carbon monoxide (CO), and ozone (O_3_) based on the adjustments made in Model 3, creating the whole model. Recognizing the potential delayed associations, we considered different lag structures, ranging from lag0 to lag2, in our analysis. To estimate cumulative associations, we used moving averages across the lag periods, extending from mv01 (moving average concentrations of lag0 and lag1) to mv02 (moving average concentrations of lag0 to lag2). The results were presented as percentage changes in blood lipid levels or FPG and the associated 95% confidence intervals (CIs) for each 1 μg/m^3^ increase in NO_2_ concentration.

The genotype–environment interaction for each trait was estimated using maximum likelihood-based variance component models implemented in SOLAR (version 9.0.0). The variance of each phenotype (δ^2^) was partitioned into two components: additive genetic factors and environmental factors. Heritability (h^2^) was defined as the ratio of the residual variance of the trait resulting from additive genetic factors (${\;\delta }_{g}{}^{2}$) compared to the total residual phenotypic variance (δ^2^). The genotype–environment interaction estimates the relative contributions of genes to trait variance across different environments [27]. The additive genetic component, as a function of environmental exposure level, has been previously defined and expressed as $\delta_{g}{}^{2}=\exp\left(\alpha_{g}+\gamma_{g}\left(e_{i}-\bar{e}\right)\right)$, where $e_{i}$ is the environmental exposure level to which individual i is exposed, while $\bar{e}$ is the average environmental exposure level, and $\alpha_{g}$ and $\gamma_{g}$ are both parameters to be estimated. If there is a genotype–environment interaction, then $\gamma_{g}$ is not equal to zero, implying that the heritability of the trait changes at different levels of an environmental exposure. All statistical tests were two-sided, and *p* < 0.05 was considered statistically significant. R 4.1.2 was used for data analysis and results output.

## 3. Results

### 3.1. Basic Characteristics

Table 1 provides an overview of the essential characteristics of the study participants. Between August 2015 and December 2017, a total of 1563 individuals, with an average age of 57.23 years, were included in our study. Of the participants, 43.57% were male. The mean concentrations of key parameters, including TG, TC, HDLC, LDLC, and FPG, were 1.75 mmol/L, 5.02 mmol/L, 1.31 mmol/L, 3.01 mmol/L, and 5.64 mmol/L, respectively.

Table 2 shows the distribution of daily NO_2_ concentrations. The NO_2_ data are matched to the date of the blood sample collection. Throughout the study period, the means of NO_2_ concentration were 27.48 μg/m^3^ (SD: 12.05 μg/m^3^), 27.45 μg/m^3^ (SD: 11.49 μg/m^3^), and 28.17 μg/m^3^ (SD: 11.18 μg/m^3^) for the day of blood draw (lag0), the previous day (lag1), and two days before the investigation (lag2), respectively. We also calculated the moving average concentrations for lag0 and lag1 (mv01) and the moving average concentrations from lag0 to lag2 (mv02), which were 27.47 μg/m^3^ (SD: 11.22 μg/m^3^) and 27.70 μg/m^3^ (SD: 10.64 μg/m^3^), respectively.

### 3.2. Association of NO_2_ with Blood Lipids and FPG

The impacts of NO_2_ concentration on lipid and FPG levels are detailed in Table 3 and Figure 1. A noteworthy reduction in HDLC was evident with rising NO_2_ levels in each lag structure (*p* < 0.05). We also identified a positive link between NO_2_ concentration and TG across most lag structures. Furthermore, an increase of 1 μg/m^3^ in NO_2_ was statistically linked to a 1.400% (0.341%, 2.470%) rise in FPG on day lag0. No significant associations were observed for TC and LDLC, although the point estimates indicated potential effects of increased NO_2_. Additionally, we noted variability in the lag effect of NO_2_ on lipids and glucose among different phenotypes. For TG, the lag effect of NO_2_ was more pronounced, showing significantly larger increments on days lag 2 and lag 1 compared to day lag 0. In contrast, with HDLC and GLU, the association with NO_2_ was most robust on day lag 0. To verify the robustness of these findings and explore potential interactions, we conducted stratified analyses based on sex, age, and BMI, which consistently upheld the results (Appendix A online, Appendix A).

### 3.3. Genotype–NO_2_ Interaction Effects on Blood Lipid Levels and FPG

The results of genotype–NO_2_ interactions are depicted in Figure 2. Even after accounting for various covariates, we identified a noteworthy genotype–NO_2_ interaction in both HDLC and FPG across all lag structures (*p* < 0.05). In the case of HDLC (Appendix A online, Appendix A) and FPG (Appendix A online, Appendix A), the additive genetic effects displayed a diminishing trend as the NO_2_ concentration increased, indicating a concurrent reduction in heritability.

## 4. Discussion

The present study provides new insights into the connections between NO_2_ and blood lipids or FPG, the critical risk factors for cardiometabolic diseases. In our investigation, elevated NO_2_ levels were related to higher TG and FPG, while also being associated with reduced HDLC. Furthermore, we discovered a potential interaction between genetic variations as a whole and NO_2_ concerning HDLC and FPG.

In partial alignment with prior research, our study unveiled a relationship between heightened NO_2_ concentrations and a reduction in HDLC. A survey conducted in urban areas of China reported that every 10 μg/m^3^ increase in NO_2_ corresponded to a 1.6% decrease in HDLC [31]. Similarly, findings from a study in South Korea produced analogous results [32]. Additionally, we established that NO_2_ levels are associated with decreased HDLC levels in Chinese adults aged 30–79 [33]. Moreover, our study suggested a positive association between NO_2_ and TG, which aligns with the findings of Yang et al., who observed a 6.0% increase in TG for every 10 μg/m^3^ rise in NO_2_ [31]. Furthermore, our study identified a positive trend in the associations of NO_2_ with LDLC and TC, although not statistically significant. These findings have generated controversy in prior studies. Evidence from a Chinese multi-ethnic cohort study demonstrated that heightened NO_2_ levels significantly elevated TC and LDLC [33]. Conversely, results from another study indicated that increased NO_2_ was linked to higher TC but not LDLC [34]. Simultaneously, we also identified a noteworthy positive relationship between NO_2_ and FPG, particularly evident in lag0 and mv02, hinting at a possible delayed effect. These observations align with previous research that has illuminated the glycemic impacts of air pollution [6,35,36,37]. Furthermore, several cohort studies have substantiated the association between NO_2_ exposure and an increased risk of diabetes [37,38].

The inconsistencies observed in these findings can be attributed to differences in study populations, air pollution levels, the duration of exposure, the accuracy of exposure, and outcome measurements. Additionally, inherent residual confounding variables in observational studies may have contributed to these variations, including confounders related to diet, physical activity, occupation, and indoor environment. Specifically, our study site is situated in a rural area, where the dietary patterns of the study population are mainly herbivorous, with lower meat consumption compared to urban populations [39,40]. People in rural areas typically burn straw, wood, or coal for cooking, as opposed to using cleaner energy sources, potentially leading to elevated NO_2_ levels in residential areas. Furthermore, residents in rural areas tend to have higher levels of physical activity, which can impact blood lipid profiles, but may also result in increased inhalation of air pollutants, leading to higher exposure levels. This complex interplay of various exposure sources and diverse lifestyles within the rural population complicates the association between NO_2_ and blood lipids. Therefore, our findings underscore the need for further exploration in the future to gain a more comprehensive understanding of these intricate relationships.

Genotype–environment interactions were also a notable focus of the current study. We observed a statistically significant interaction between genetic effects and NO_2_. The overall additive genetic effect gradually diminished as NO_2_ concentrations increased, signifying a concurrent reduction in the influence of genetic factors. This phenomenon may be attributed to the cumulative and escalating impact of environmental exposure as NO_2_ concentrations rise, a trend that has been substantiated in previous research [41]. Our findings underscored the importance of future studies that concentrate on identifying specific genetic variants that play a role in gene–environment interactions, particularly regarding the exposure to NO_2_ and its effects on blood lipids or FPG.

The underlying biological mechanism through which air pollution influences blood lipids and FPG remains a subject of ongoing investigation. A prevailing hypothesis suggests that air pollutants induce changes in blood lipids and FPG by inciting oxidative stress and triggering systemic inflammation [38,42,43]. These alterations in oxidative stress are particularly pronounced in the case of HDLC [44]. The inflammatory hypothesis regarding the impact of air pollution on FPG primarily targets adipose tissue [6,38]. Inflammation within adipose tissue can lead to an increased pro-inflammatory-to-anti-inflammatory macrophage ratio, potentially disrupting insulin signaling and contributing to insulin resistance [45]. Another potential pathway through which changes in FPG may occur is that air pollution directly affects insulin resistance [46]. Likewise, there might be pathways linked to DNA methylation that enable air pollution to influence blood lipid levels. Prior studies have established connections between air pollutants and DNA methylation levels in specific genes associated with lipid metabolism and inflammatory responses [47,48,49]. These findings partially elucidate the results of the genotype–NO_2_ interaction. Nonetheless, the precise mechanism necessitates further exploration.

To the best of our knowledge, the current study is the first examining genotype–NO_2_ interactions for cardiometabolic traits to be reported, which provided clues to identify underlying genetic variants in blood lipids and FPG. In addition, the use of a pedigree-based design was also a strength. Owing to the Tulous’ unique architectural forms, special culture, and remote geographical location, native residents are isolated from other ethnic populations, resulting in their homogenous genetic backgrounds. Therefore, the population stratification will be better controlled in this pedigree-based cohort. This study also has some limitations. First, although we used air pollution data from a few days prior to explore the short-term effects, it was still a cross-sectional study with weak causal inference ability. Second, we used fixed-site air pollution monitoring data rather than individual exposures, as detailed home address information was not available, which might have led to some degree of measurement error. Third, although we adjusted for a number of confounders, residual confounding from unknown and unmeasured sources might be present. Finally, the study was based on a population from the Tulou pedigrees in southern China, which might have affected the generalizability of the results to other populations, yet the internal validity of the study remains significant. In the future, the ongoing prospective follow-up and expansion of the Fujian Tulou Pedigree-based Cohort Study will enable us to examine the long-term effect of nitrogen dioxide on blood lipids and glucose levels. Building on the current findings, we aspire to conduct comprehensive genetic investigations to unravel the interactions between specific genes and the nitrogen dioxide exposure. This endeavor aims to provide valuable insights for more precise strategies for the prevention and management of cardiovascular diseases in the years to come.

## 5. Conclusions

In conclusion, this study revealed significant associations between NO_2_ exposure and alterations in blood lipid profiles and glucose levels, with NO_2_ being associated with elevated TG and FPG, as well as reduced HDLC. Furthermore, the results highlighted the importance of considering genotype–environment interaction effects involving genetic factors and NO_2_ exposure on blood lipid and FPG levels. The presence of genotype–NO_2_ interactions provided the basis for subsequent studies focusing on analyzing the interaction effects of specific genetic loci with NO_2_ and helped to explain the results of previously published studies.

## Figures and Tables

**Figure 1 biology-12-01470-f001:**
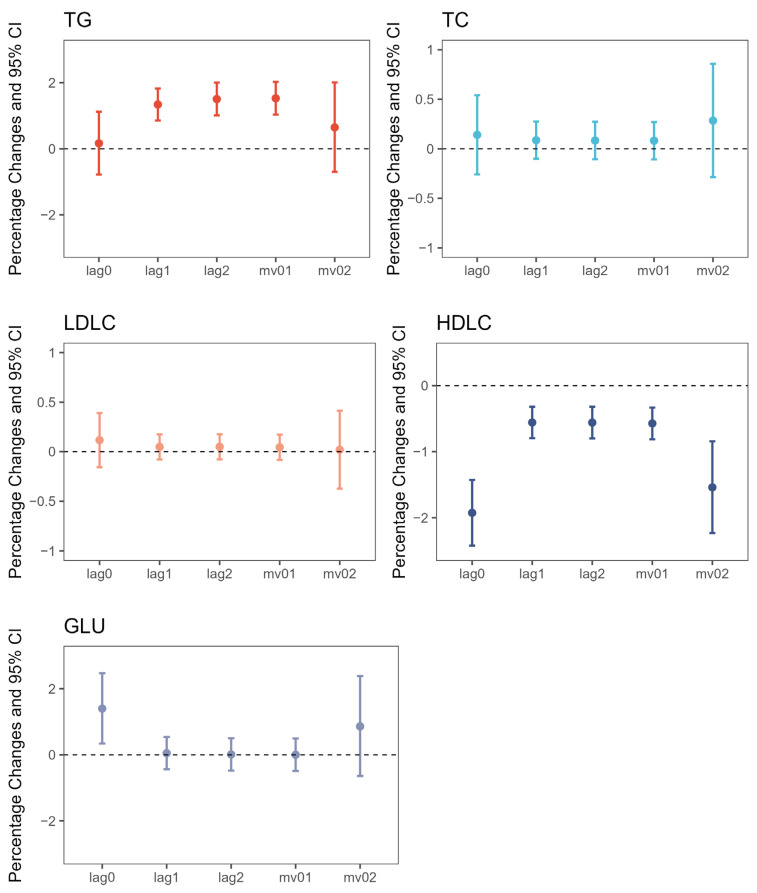
Percentage changes with 95% confidence intervals (95% CIs) in blood lipids and FPG associated with per 1 μg/m^3^ increase in NO_2_ of different lag structures. Multivariable model was adjusted for age, sex, BMI, education level, family ID as a random effect term, physical activity, smoking, drinking, vegetable and fruit intake, whole-grain intake, meat intake, tea intake, the use of antihypertensive/antidiabetic/antihyperlipidemic drugs, and concentration of PM_2_._5_, PM_10_, CO, and O_3_. HDLC, high-density lipoprotein cholesterol; Lag0, concentrations on the day of investigation; Lag1, concentrations on the day previous to investigation; Lag2, concentrations two days before investigation; LDLC, low-density lipoprotein cholesterol; Mv01, moving average concentrations of Lag0 and Lag1; Mv02, moving average concentrations of Lag0 to Lag2; TC, total cholesterol; TG, triglycerides.

**Figure 2 biology-12-01470-f002:**
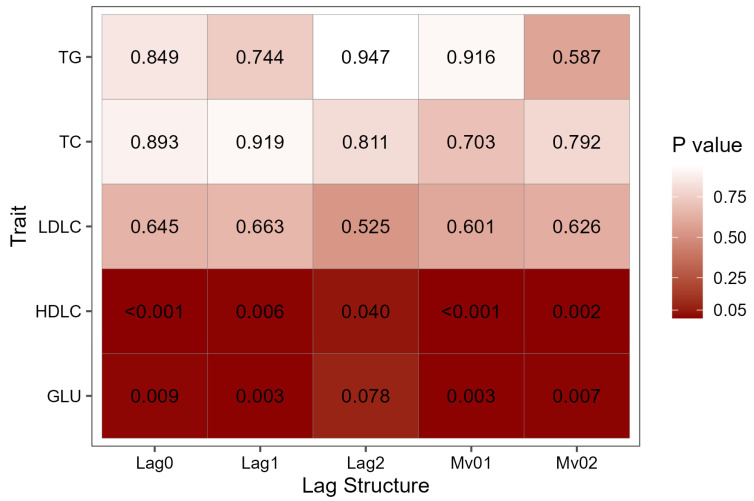
Interactions effects of genotype with NO_2_ on blood lipid levels and fasting plasma glucose in different lag structures. Significant results are indicated with dark red patches and *p* values (*p* < 0.05). HDLC, high-density lipoprotein cholesterol; Lag0, concentrations on the day of investigation; Lag1, concentrations on the day previous to investigation; Lag2, concentrations two days before investigation; LDLC, low-density lipoprotein cholesterol; Mv01, moving average concentrations of Lag0 and Lag1; Mv02, moving average concentrations of Lag0 to Lag2; TC, total cholesterol; TG, triglycerides.

**Table 1 biology-12-01470-t001:** Characteristics of study participants.

Characteristic		Total	Male	Female
N (%)		1563 (100.00)	681 (43.57)	882 (56.43)
Age, mean (SD), y		57.23 (13.24)	58.38 (12.97)	56.35 (13.39)
BMI (kg/m^2^)		23.35 (3.36)	23.40 (3.48)	23.30 (3.26)
Smoking (%)	Never	1161 (74.3)	304 (44.6)	857 (97.2)
	Current	305 (19.5)	283 (41.6)	22 (2.5)
	Previous	97 (6.2)	94 (13.8)	3 (0.3)
Alcohol drinking (%)	Never	1360 (87.0)	513 (75.3)	847 (96.0)
	Current	166 (10.6)	137 (20.1)	29 (3.3)
	Previous	37 (2.4)	31 (4.6)	6 (0.7)
Physical activity, mean (SD), MET-h/week		63.34 (91.66)	67.00 (99.95)	60.51 (84.67)
Vegetable intake, mean (SD), g/d		228.00 (143.08)	235.04 (156.41)	222.56 (131.70)
Fruit intake, mean (SD), g/d		46.26 (67.59)	41.74 (62.92)	49.75 (70.83)
Meat intake, mean (SD), g/d		101.95 (71.72)	112.95 (82.02)	93.45 (61.33)
Whole-grain intake, mean (SD), g/d		11.12 (20.19)	10.43 (20.60)	11.66 (19.87)
TG, mean (SD), mmol/L		1.75 (1.61)	1.93 (1.82)	1.61 (1.41)
TC mean (SD), mmol/L		5.02 (1.89)	4.81 (1.10)	5.18 (2.30)
HDLC, mean (SD), mmol/L		1.31 (0.60)	1.20 (0.66)	1.40 (0.53)
LDLC, mean (SD), mmol/L		3.01 (0.81)	2.96 (0.82)	3.05 (0.80)
FPG, mean (SD), mmol/L		5.64 (1.41)	5.81 (1.65)	5.50 (1.17)

BMI, body mass index; FPG, fasting plasma glucose; HDLC, high-density lipoprotein cholesterol; LDLC, low-density lipoprotein cholesterol; SD, standard deviation; TC, total cholesterol; TG, triglycerides.

**Table 2 biology-12-01470-t002:** Summary statistics for NO_2_ concentrations (μg/m^3^).

Lag Structures	Mean ± SD	Minimum	Percentile			Maximum	IQR
			25th	50th	75th		
Lag0	27.48 ± 12.05	8.00	18.00	26.00	33.00	68.00	15.00
Lag1	27.45 ± 11.49	6.00	19.00	26.00	33.00	68.00	14.00
Lag2	28.17 ± 11.18	9.00	20.00	26.00	35.00	68.00	15.00
Mv01	27.47 ± 11.22	7.00	19.00	27.50	31.00	65.00	12.00
Mv02	27.70 ± 10.64	7.67	19.33	29.33	31.50	65.00	12.17

IQR, interquartile range; Lag0, concentrations on the day of investigation; Lag1, concentrations on day previous to investigation; Lag2, concentrations two days before investigation; Mv01, moving average concentrations of Lag0 and Lag1; Mv02, moving average concentrations of Lag0 to Lag2; SD, standard deviation.

**Table 3 biology-12-01470-t003:** Percentage changes with 95% confidence intervals (95% CIs) in TG, TC, LDLC, HDLC, and FPG associated with per 1 μg/m^3^ increase in NO_2_ for different lag structures.

Trait	Lag Structure	Model 1	Model 2	Model 3	Model 4
TG	Lag0	**1.085 (0.634, 1.537)**	**1.630 (1.149, 2.114)**	**1.093 (0.657, 1.530)**	0.166 (−0.780, 1.121)
	Lag1	**1.175 (0.716, 1.636)**	**1.660 (1.175, 2.146)**	0.711 (−0.122, 1.551)	**1.340 (0.857, 1.826)**
	Lag2	**1.187 (0.727, 1.649)**	0.671 (−0.086, 1.433)	**1.394 (0.910, 1.881)**	**1.505 (1.011, 2.001)**
	Mv01	0.098 (−0.574, 0.774)	0.963 (0.535, 1.393)	**1.568 (1.071, 2.068)**	**1.528 (1.032, 2.027)**
	Mv02	**1.424 (0.954, 1.897)**	**1.076 (0.642, 1.511)**	**1.591 (1.092, 2.093)**	0.646 (−0.698, 2.008)
TC	Lag0	0.011 (−0.154, 0.177)	0.107 (−0.067, 0.283)	0.101 (−0.075, 0.277)	0.140 (−0.259, 0.541)
	Lag1	0.011 (−0.156, 0.178)	0.101 (−0.073, 0.276)	0.123 (−0.223, 0.470)	0.086 (−0.101, 0.274)
	Lag2	0.006 (−0.160, 0.173)	0.160 (−0.142, 0.462)	0.065 (−0.113, 0.244)	0.084 (−0.105, 0.272)
	Mv01	0.214 (−0.071, 0.500)	0.100 (−0.075, 0.274)	0.064 (−0.115, 0.244)	0.081 (−0.107, 0.270)
	Mv02	0.110 (−0.064, 0.284)	0.095 (−0.081, 0.271)	0.058 (−0.121, 0.237)	0.284 (−0.286, 0.858)
LDLC	Lag0	0.108 (−0.004, 0.220)	0.002 (−0.116, 0.119)	0.022 (−0.098, 0.141)	0.116 (−0.157, 0.390)
	Lag1	0.097 (−0.015, 0.210)	0.006 (−0.112, 0.124)	0.196 (−0.040, 0.432)	0.048 (−0.079, 0.175)
	Lag2	0.093 (−0.020, 0.205)	0.159 (−0.046, 0.365)	0.058 (−0.062, 0.179)	0.049 (−0.078, 0.176)
	Mv01	**0.221 (0.025, 0.417)**	0.014 (−0.105, 0.134)	0.054 (−0.067, 0.175)	0.044 (−0.083, 0.172)
	Mv02	0.005 (−0.113,0.123)	0.025 (−0.094, 0.145)	0.049 (−0.072, 0.171)	0.020 (−0.373, 0.414)
HDLC	Lag0	**−0.680 (−0.898, −0.461)**	**−0.363 (−0.579, −0.146)**	**−0.369 (−0.585, −0.153)**	**−1.926 (−2.421, −1.428)**
	Lag1	**−0.674 (−0.894, −0.453)**	**−0.371 (−0.587, −0.155)**	**−0.479 (−0.881, −0.076)**	**−0.558 (−0.796, −0.320)**
	Lag2	**−0.683 (−0.904, −0.463)**	**−0.872 (−1.261, −0.481)**	**−0.577 (−0.806, −0.347)**	**−0.560 (−0.799, −0.320)**
	Mv01	**−1.896 (−2.275, −1.516)**	**−0.340 (−0.553, −0.126)**	**−0.573 (−0.803, −0.342)**	**−0.573 (−0.812, −0.332)**
	Mv02	**−0.364 (−0.579, −0.148)**	**−0.354 (−0.569, −0.139)**	**−0.582 (−0.813, −0.352)**	**−1.540 (−2.232, −0.843)**
GLU	Lag0	0.331 (−0.104, 0.768)	0.117 (−0.334, 0.570)	0.158 (−0.303, 0.621)	1.400 (0.341, 2.470)
	Lag1	0.274 (−0.165, 0.715)	0.126 (−0.326, 0.580)	0.798 (−0.114, 1.718)	0.049 (−0.437, 0.538)
	Lag2	0.265 (−0.174, 0.706)	0.080 (−0.693, 0.859)	0.143 (−0.321, 0.609)	0.010 (−0.480, 0.503)
	Mv01	1.163 (0.403, 1.929)	0.147 (−0.309, 0.604)	0.093 (−0.374, 0.562)	0.001 (−0.490, 0.494)
	Mv02	0.085 (−0.362, 0.535)	0.153 (−0.307, 0.615)	0.083 (−0.385, 0.553)	0.861 (−0.641, 2.386)

HDLC, high-density lipoprotein cholesterol; Lag0, concentrations on the day of investigation; Lag1, concentrations on the day previous to investigation; Lag2, concentrations two days before investigation; LDLC, low-density lipoprotein cholesterol; Mv01, moving average concentrations of Lag0 and Lag1; Mv02, moving average concentrations of Lag0 to Lag2; TC, total cholesterol; TG, triglycerides. Model 1 adjusted for age, sex, BMI, education level, and family ID; Model 2 additionally adjusted for physical activity, smoking, drinking, vegetable and fruit intake, whole-grain intake, meat intake, and tea consumption on the basis of model 1; Model 3 additionally adjusted for use of antihypertensive/antidiabetic/antihyperlipidemic drugs on the basis of model 2; Model 4 additionally adjusted for concentration of PM_2_._5_, PM_10_, CO, and O_3_. Significant results are indicated in bold (*p* < 0.05).

## Data Availability

The data used in this study are available from the corresponding authors upon reasonable request.

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
