# Peer review of "Short-Term Exposure to Nitrogen Dioxide Modifies Genetic Predisposition in Blood Lipid and Fasting Plasma Glucose: A Pedigree-Based Study"

_biology, 2023, doi:10.3390/biology12121470_

Round 1
Reviewer 1 Report
Comments and Suggestions for Authors
Abstract section should not be in number form. Please revise and integrate the abstract into one paragraph.
Introduction section should include a state-of-the-art literature review, in which 22 references may not be adequate. Please consider increasing both the length and the work cited in this section.
The materials and methods section are clear.
The results section should not be a separate section, but in a same section with the discussion, namely result and discussion.
Since this study involves human, has this research go through ethical review process?
Also, is the personal information collected from this study been protected?
It is important to hide the personal identity of these individuals participate in this study.
Why are the locations of air quality monitoring stations not shown in this section?
It should be clearly represented in a map. Please add the map of the air quality monitoring system.
It is also important to add the map of study area where this study is conducted. Since Fujian province is a big province, it is essential to include the exact location in a map (such as longitude and latitude).
The conclusion section is too short and did not summarized the finding of this manuscript. Please revised this section.
If there are no patents applied for this study, please remove this section.
Comments on the Quality of English LanguagePlease check the spelling and grammar throughout the text.
Reviewer 2 Report
Comments and Suggestions for Authors
This is a report aiming to investigate the correlation between short-term NO2 exposure and blood lipids, fasting plasma glucose, as well as the potential interactive effects with genetic factors. The report would benefit from several improvements in its English language and content. The findings are intriguing, but the review should consider the following points:
Provide Background on NO2 Source: It would be helpful to include some background information about the sources of NO2, giving readers context about where this pollutant originates.
Correct Reference for NO2 Impact: In lines 67-68, the report mentions that "NO2 contributes to cardiovascular events by causing abnormalities in lipid and glucose metabolism." However, references 7-9 may not be the most appropriate sources for this statement. Please review and amend this accordingly.
Examine Participant Characteristics: Table 1 outlines the characteristics of the study participants. It would be beneficial to discuss whether these characteristics play a role in blood lipids, fasting plasma glucose, genetic factors, and how NO2 impacts them.
Consider Other Air Pollutants: Since this study solely measures NO2, it's important to address the potential influence of other air pollutants. It's challenging to separate the effects of different pollutants on human health.
Mentioning the measurement of pollutants like PM10, PM2.5, CO, and discussing their associations would provide a more comprehensive perspective. The report should not isolate NO2 from the broader context of air pollution.
Clarify Figure 1: In Figure 1, which shows "Percentage changes with 95% confidence intervals (95% CIs) in blood lipids and FPG associated with a per 1 μg/m3 increase in NO2 of different lag structures," please clarify the differences and changes being represented.
Discuss Future Directions: It would be valuable to include a section about the authors' future plans based on the current findings. What further research or actions do they intend to pursue as a result of these results?
Revising the report with these considerations in mind will enhance its clarity and completeness.
Comments on the Quality of English LanguageThe report would benefit from several improvements in its English language and content.
Round 2
Reviewer 1 Report
Comments and Suggestions for Authors
As shown in section 2.2 Air pollutant data:
We acquired air pollutant data, including the 24-hour average concentrations of NO2, 409 Particulate Matter (PM2.5 and PM10), Carbon monoxide (CO), and Ozone (O3) in Zhang- 410 zhou City, Fujian Province, spanning from January 1, 2015, to December 31, 2017. These 411 data were sourced from the National Air Pollution Monitoring System. The national 412 standard monitoring station is equipped with 24-hour automatic monitoring instruments 413 for NO2 concentrations. Daily average concentrations were computed for our analysis. It 414 is noteworthy that the monitoring data has been demonstrated to be a reliable represen- 415 tation of population exposure to air pollution in China [23, 24]. A map illustrating the site 416 of the Fujian Tulou Pedigree-based Cohort study and the locations of air pollutant moni- 417 tors can be found in the Supplementary Material online, Figure S1.
Why is the air quality data used not being the most recent? Can the author explain the reason? The data used in this study is starting from 2015, which is almost a decade ago. Should more recent data be included in this study? Because the overall air quality levels in China has been improving, including the decreasing levels for nitrogen dioxide (NO2), as the higher adoption rate of electric vehicles (EVs) in the transportation sector and the transition to renewable energy for the energy sector.
There are also other air pollutants such as PM, CO, SO2, and O3, why are these pollutants not studied for the health impact on the blood lipid and fasting plasma glucose? Because it is known that there are six air pollutants (PM10, PM2.5, CO, NO2, SO2, and O3) made up the air quality index (AQI). So, the authors must justify why only NO2 was studied while the others were not considered.
Also, is it a usual practice to have three joint first authors and two corresponding authors in a scientific paper? I’m not sure if this is ok. Can the authors please provide a justification?
Nevertheless, the conclusion section remained to be subpar and further improvements must be made before consideration for publication.
Comments on the Quality of English Language
The authors should revised the spelling and grammar throughout the manuscript.
Author Response
Thank you for your comments! Please see the attachment for our responses.

Reviewer 2 Report
Comments and Suggestions for Authors
Manuscript much improved.
Comments on the Quality of English LanguagePlease adjust some of the expression. For example, lines 112-113 "Previous research has concluded that lipid and blood glucose levels could explain the relationship between air pollution and CVD." It is not clear how the above conclusion was made and what's the link to the following part about the NO2. The following part used separate references as evidence.
Author Response

(The authors gave the same response as above.)

Round 3
Reviewer 1 Report
Comments and Suggestions for Authors
Some improvements have been made to the manuscript and the authors have answered the questions raised by the reviewer. Thus, I have no further objection for the manuscript to be published.
Comments on the Quality of English LanguageOk.